# Latinx and Indigenous Mexican Caregivers’ Perspectives of the Salton Sea Environment on Children’s Asthma, Respiratory Health, and Co-Presenting Health Conditions

**DOI:** 10.3390/ijerph20116023

**Published:** 2023-06-01

**Authors:** Ann Marie Cheney, Gabriela Ortiz, Ashley Trinidad, Sophia Rodriguez, Ashley Moran, Andrea Gonzalez, Jaír Chavez, María Pozar

**Affiliations:** 1Department of Social Medicine Population and Public Health, School of Medicine, University of California Riverside, Riverside, CA 92521, USA; 2Department of Anthropology, University of California Riverside, Riverside, CA 92521, USA; 3College of Natural & Agricultural Sciences, University of California Riverside, Riverside, CA 92521, USA; 4David Geffen School of Medicine, University of California Los Angeles, Los Angeles, CA 90095, USA; 5Conchita Servicios de la Comunidad, Mecca, CA 92254, USA

**Keywords:** asthma, child health, respiratory conditions, environmental health, Indigenous Mexicans, Latinx health, Purépecha, Salton Sea

## Abstract

This research investigated Latinx and Indigenous Mexican caregivers’ perspectives of the Salton Sea’s environment (e.g., dust concentrations and other toxins) on child health conditions. The Salton Sea is a highly saline drying lakebed located in the Inland Southern California desert borderland region and is surrounded by agricultural fields. Children of Latinx and Indigenous Mexican immigrant families are especially vulnerable to the Salton Sea’s environmental impact on chronic health conditions due to their proximity to the Salton Sea and structural vulnerability. From September 2020 to February 2021, we conducted semi-structured interviews and focus groups with a total of 36 Latinx and Indigenous Mexican caregivers of children with asthma or respiratory distress living along the Salton Sea. A community investigator trained in qualitative research conducted interviews in Spanish or Purépecha, an indigenous language spoken by immigrants from Michoacán, Mexico. Template and matrix analysis was used to identify themes and patterns across interviews and focus groups. Participants characterized the Salton Sea’s environment as toxic, marked by exposure to sulfuric smells, dust storms, chemicals, and fires, all of which contribute to children’s chronic health conditions (e.g., respiratory illnesses such as asthma, bronchitis, and pneumonia, co-presenting with allergies and nosebleeds). The findings have important environmental public health significance for structurally vulnerable child populations in the United States and globally.

## 1. Introduction

Asthma is the most common chronic condition among children in the United States (US), with ~11.3% of the US child population estimated to suffer from asthma) [1]. Yet, children’s risk for asthma varies depending on social status (e.g., race/ethnicity, indigeneity, citizenship), economic factors, and environmental exposures. Childhood asthma rates are highest among racial/ethnic minority children [2], who are often from low-income families living in poor neighborhoods near busy highways and industrial zones [3,4]. Low-income and racial/ethnic minority children, when compared to middle-class white children, experience a lower quality of life due to higher exposure to environmental hazards [5,6]. While geography matters, children in both rural and urban communities are exposed to harmful environmental hazards. For instance, particulates from neighborhoods in urban settings are often from fossil fuels emitted through sources such as congested traffic and construction sites. Particulates from neighborhoods in rural settings are often from windblown dust from fields or unpaved roads and fossil fuels from agricultural equipment. 

Studies have shown that racial/ethnic minority children, including Black/African American, Latinx/Hispanic, and first- and second-generation immigrant children, are disproportionally exposed to particulate matter (solids or liquids in the air) with diameters of 2.5 microns or less (PM_2.5_) which include smoke from fires and emissions from industrial facilities among other sources [3,7]. Fine particles are especially dangerous because their small size can more easily enter children’s lungs and potentially the bloodstream, contributing to poor health outcomes [8]. 

Structural and social determinants of health (SDOH) play a critical role in children’s risk for asthma and related respiratory health conditions. SDOH, the conditions into which individuals are born, grow up, live, work, and age, act as key determinants of health. They determine access to education, quality housing, and safe neighborhoods, as well as exposure to environmental hazards such as poor air quality [9]. Yet, we also know that historical processes of colonialism, classism, and racism have historically set up inequities within institutions and social life, patterning the distribution of SDOH in minoritized populations and contributing to health inequities [10]. As Brewer and colleagues argue, childhood asthma rates are the embodiment of environmental hazards, revealing the social pattern of inequity and the disadvantage of racial/ethnic minority children [11]. 

A unique example of the confluence of structural and SDOH on childhood asthma is among low-income Latinx and Indigenous Mexican children in communities bordering the Salton Sea in the desert region of Inland Southern California [12,13]. The prevalence of childhood asthma among children living along the southern part of the Salton Sea is 20–22.4% [14], double the state and national childhood asthma prevalence of 10% and 11.3%, respectively [1]. The air quality around the Salton is a significant local, regional, and statewide concern, and there are efforts to restore the ecosystem around the Salton Sea (e.g., 10-year Salton Sea plan) [15] and improve the region’s air quality through resources and funds that align with the designation of an AB 617 community [16]. 

In this article, we build on our existing work on the working and living conditions that contribute to health disparities among the Latinx and Purépecha farm working communities along the northern part of the Salton Sea [12,17] and consider the impact of the Salton Sea environment on the health and wellbeing of children living along its borders. The funded project, Childhood Asthma and the Salton Sea, is part of a larger National Institute for Minority Health and Health Disparities-funded project focused on exposure to aerosolized environmental contaminants from the Salton Sea’s drying lakebed. One important component of the study is the engagement of caregivers and key stakeholders in partnered public health research focused on understanding the impacts of the Salton Sea on children’s respiratory health. This study uniquely focuses on caregivers’ understanding of the Salton Sea’s impact on the health of children diagnosed with asthma or chronic respiratory health problems through qualitative interviews and testimonials provided by Latinx and Indigenous Mexican caregivers of children with asthma and respiratory problems.

## 2. Methods

The complete study was carried out from fall 2019 to spring 2021. Qualitative interviews were conducted from September 2020 to February 2021. We used principles of community-based participatory research (CBPR) whereby decision-making was collaborative, knowledge was co-created, and resources were shared [18]. In line with CBPR approaches, at the start of the project, we convened a community advisory board (CAB) of 12 members representing parents of children with asthma, environmental justice organizations, healthcare systems, and community health workers (CHWs) or promotores de salud. The advisory board met quarterly to guide project activity. CAB members reviewed interview guides and recruitment material and provided input on recruitment, initial findings, and community dissemination strategies. 

Prior to the start of the research, we obtained ethical approval from the University of California, Riverside Institutional Review Board. All participants provided electronic consent prior to the start of data collection. 

### 2.1. Setting 

Our study focused on childhood asthma among Latinx and Indigenous Mexican children in the rural desert region of Inland Southern California with a focus on the Eastern Coachella Valley which includes the unincorporated communities of Thermal, North Shore, Oasis, and Mecca (see Figure 1 for a map of the study region). These unincorporated communities do not have their own government structure and are characterized by poor housing and water infrastructure exposing residents to environmental hazards [13]. These communities reside on the northern part of the Salton Sea, a once booming resort area that swiftly changed to low-income housing for immigrant communities as the agricultural runoff, the main source of water, increased the lake’s salinity contributing to disease and infection among the lake’s habitat [19,20]. 

#### 2.1.1. The Salton Sea

The Salton Sea occupies the prehistoric lakebed of Lake Cahuilla located in southeast California between Riverside and Imperial Counties and in the US-Mexico borderlands (approximately 90 miles from the US-Mexico border, see Figure 1). It came into being in the early 1900s due to an error with the rerouting of overflow from the Colorado River to the Imperial Valley. Once the overflow was controlled, the resulting lake was utilized for draining agricultural runoff from nearby farmlands, which has created a highly toxic body of water, further exacerbated by rising temperatures in the area [20] that have contributed to water evaporation, which exposes the sediment where the toxins (e.g., heavy metals, bacteria, and pesticides) rest. Water politics, involving the 2003 Color Riverside Agreement and recent negotiations by the Biden Administration, have significantly reduced water to the Imperial Valley and its agricultural lands, which was a significant source of water for the Salton Sea [21]. The result is a rapidly shrinking lake and exposed lakebed. 

This environment has harmful effects on wildlife (fish and migrating birds) and on the human population around the Salton Sea [22]. Communities surrounding the Salton Sea have a higher incidence of respiratory distress, especially among children [23], compared to the general population [24]. Recent studies using environmental chamber models to simulate the Salton Sea found that daily exposure to the Salton Sea’s aerosols induced non-allergic inflammatory responses in animal models [25]. The activation of non-allergic inflammation genes upon chronic exposure to the lake’s aerosols likely results in lung inflammation and affects the lung health of those living along its borders. 

#### 2.1.2. Structurally Vulnerable Populations Living along the Salton Sea

This region is home to a large Latinx and Indigenous Mexican immigrant farm working population that lives in low-income and poverty-stricken communities (median household income in Mecca, one of the communities at the northern part of the Salton Sea, is $25,202) [26] and works in the nearby agricultural fields. This population experiences significant disparities in health due to their structurally vulnerable positions and chronic daily exposure to stressors that compromise their physical and mental health [13]. We draw on the concept of structural vulnerability in the anthropological and social medicine literature to illustrate how positionality places Latinx and Indigenous Latin American farm-working populations in the US in precarious social positions within a hierarchical social order and power relations that expose them to structural violence. This term (i.e., structural violence) refers to a violence that is often invisible and plays out in seemingly ordinary ways (e.g., healthcare insurance requires documentation status) [27,28]. 

The adult population living along the Salton Sea’s border is predominantly an immigrant, mono-lingual Spanish-speaking Latinx population born in Mexico that travelled to the region to work in the agricultural fields, whereas the majority of the child population was born in the US and is of Mexican heritage. Furthermore, among this population is one of the largest Purépecha communities in the US, an Indigenous group from the Mexican state of Michoacán. There are an estimated 6000 to 10,000 Purépecha-identifying individuals in this region of the desert and most are from Ocumicho. Many Purépecha are monolingual and speak their traditional language limiting their understanding of Spanish or English and excluding them from Salton Sea decision making [29]. As argued elsewhere, this Indigenous Mexican child population is especially vulnerable to environmental racism and exposed to poor air quality affecting their respiratory health [30]. Additionally, the Salton Sea’s border is home to the Torres Martinez Desert Cahuilla Indians. While these lands present safety for undocumented immigrants, especially members of the Purépecha community, they are also sites of economic abuses (e.g., inflated rent) and environmental hazards (illegal dumping and fires) [13,31].

These Latinx and Purépecha populations are vulnerable to structural inequities in health due to their race/ethnicity, immigration status, indigeneity, and geographic locale in the US-Mexico borderlands. 

#### 2.1.3. Participant Recruitment

We used convenience (nonrandomized) and snowball sampling to recruit participants into the study. Snowball sampling, a variant of chain sampling, permitted the community investigator and CAB members to reach out through their social and professional networks to share study information and recruit participants into the study. Eligible participants had to be: (1) 18 years or older, (2) a caregiver of a child with asthma symptoms, which could include an official diagnosis or the prescription of asthma medication (e.g., albuterol), (3) live in a community along the Salton Sea (i.e., North Shore, Oasis, Thermal, Mecca, Desert Shore, Salton City), and (4) speak English, Spanish, and/or Purépecha.

#### 2.1.4. Qualitative and Sociodemographic Survey Data Collection

Between September and February 2021, we conducted four focus groups, three in Spanish and one in Purépecha with 16 participants, with a range of 3 to 5 participants per focus group, and 20 one-on-one interviews, 14 in Spanish and 6 in Purépecha. We followed recommendations for obtaining data saturation for focus groups, which can be achieved within 2 to 3 focus groups [32], and 12–15 semi-structured interviews with fairly homogenous samples [33].

The community investigator conducted all data collection in either Spanish or Purépecha. A semi-structured interview guide was used for both focus groups and interviews. The guide elicited information on the following topics: childhood asthma and related chronic health conditions (nosebleeds, allergies), perceptions of air quality (Salton Sea dust) and child health, perceptions of climate change and asthma and related health symptoms, healthcare services use, and home health remedies. Focus groups were the primary method of data collection and were conducted via Zoom video conferencing; however, if participants were not able to access Zoom, they could opt to participate in the research via a one-on-one interview by telephone. Focus groups lasted approximately 90 min and one-on-one interviews 30–60 min. All interviews (group or one-on-one) were audio recorded and transcribed. 

Immediately following qualitative data collection, participants were asked to complete a brief sociodemographic survey. Participants could choose to self-administer the survey using a link to a Qualtrics (online) survey or have a team member administer the survey to them. The survey collected basic socio-demographic data (age, gender, race/ethnicity, language, country of origin, employment, and education), relationship to the focal child with asthma (e.g., mother, father, grandparent, or aunt/uncle), number of children and number of those with asthma, housing type, and proximity to the Salton Sea (distance and crossroads). Basic characteristics of caregivers’ focal child were also collected, including age, gender, country of origin, language, number of years living near the Salton Sea, and overall health status. Participants received a $20 gift card in appreciation of their time and for sharing their experiences. 

#### 2.1.5. Data Analysis

The textual data from focus groups and one-on-one interviews were analyzed as one dataset. A rapid analytic approach using summary templates and matrix analysis was used to analyze the textual data [34]. When engaging marginalized communities in research, we have found this approach to be cost-effective, accessible as it does not require the purchase of software, and engaging as non-academic experts develop skills in analyzing and interpreting qualitative data [35,36]. Team members read each transcript line by line and inserted textual data from the transcripts into the template, developing a summary of responses to the interview questions. The summary also included quotes from the interviews and memos or analytic thoughts. A matrix (focus group/interview ^x^ interview topic) was created, and team members inserted condensed and simplified data from each template into the matrix. By using a template and matrix analysis approach, we engaged in an iterative process of theme identification. This approach permitted us to synthesize and then organize the textual data via the matrix to compare content across interviews (group and one-on-one) and identify patterns and emerging themes. This is a common approach used in applied health services and public health research [37]. Below we describe the patterns that emerged from our analysis of the qualitative data and use quotes in the body of the article and tables as evidence of analytic categories and emergent themes. 

## 3. Results

### 3.1. Participant Characteristics

A total of 36 caregivers participated in the study and 33 completed the socio-demographic survey. All caregivers and children had lived in communities along the northern part of the Salton Sea for at least six months. As indicated in Table 1, most caregivers were the focal child’s mother and were born in Mexico from various states, including Michoacán, Baja California, Veracruz, Sinaloa, Sonora, Tamaulipas, Guerrero, and the capital of Mexico City. All but one of the focal children (i.e., children with asthma or respiratory distress) of caregivers in the study were born in the US. Most children were born in and grew up in a community along the Salton Sea. 

### 3.2. An Overview: The Salton Sea Environment

Across the interviews, participants described the Salton Sea environment as harmful to children’s health (see Table 2). They explained that children are exposed to sulfuric smells emitted from the Salton Sea, dust storms, agricultural chemicals, and fires. They described the air as polluted and harmful to the children who breathe it. The following quote illustrates well children’s exposure to the effects of the Salton Sea environment: 

“[The schools] are very, very close to the lake, so when it’s hot and windy, the smell, and well, all the contamination rises and the children breath it … as you [the interviewer] recall, the fire that happened not long ago? Because all of that affected the kids. My daughter said to me: ‘I want to go back to school now’. But, well, it couldn’t happen … the smoke from the fire was really strong, a situation that in my point of view, it is exceeding or exceeding all [air quality] limits”.

Toxic smells. Caregivers shared that during the hot summer months of June, July, and August, the Salton Sea emits sulfuric smells that affect the respiratory health of children. Toxic smells are especially persistent and frequent in the summer months. They described the smell as one of sulfur and breathing the smell as suffocating. A mother shared: “My daughter, who has asthma tells me that she feels like she’s choking when she smells that odor [from the Salton Sea]. She tells me: ‘I feel really sick, my chest hurts when the [Salton] Sea smells like that’”. Continuing to discuss her daughter’s experience, this mother explained: “[She] tells me that she feels like she is drowning when she smells that odor”. Another mother shared a similar experience of the effects of the Salton Sea’s smell on her son’s health. “Every time you smell it [the Salton Sea] … the pollution there … sometimes he’s even incapable of breathing”.

Dust storms. During interviews, caregivers talked about the extreme weather events, specifically dust storms, that are commonplace and may occur over multiple days and are pronounced during the hot summer months. During these dust storms children often experience increased respiratory symptoms and allergies, such as irritated and watery eyes. As this mother shared: “When it is windy and there is a lot of dust, getting an [asthma] attack or having allergies is really high. It is very difficult to breathe”. 

Chemicals. Caregivers discussed children’s exposure to agricultural chemicals from the nearby agricultural fields. Children are exposed by the proximity to the field as well as household members who work in the fields and bring chemicals into the homes via their work clothes. They discussed that children in trailer parks, which are located in close proximity to the fields, are especially vulnerable: “There are a lot of pesticides around the communities where they [children] live. Trailer parks—most of the trailer parks are near agricultural fields”. 

Additionally, participants discussed the harm of local agricultural practices in which growers dump chemicals into the Salton Sea: “The growers are unloading their planters, everything goes there. We don’t know how much they clean the water. But I think there are studies in which they say there are quite a lot of chemicals in the water”. This participant continued to explain why such agricultural practices are problematic for respiratory health: 

“It [the Salton Sea] is drying up because they no longer supply it with the water that they used to supply it with. Everything [agricultural toxins] is left on the lake’s shores. When it’s windy, all this dust goes into our lungs”.

Fires. Additionally, participants talked about the burning of garbage on nearby tribal lands and its effects on air quality. In recent years, fires have occurred near schools: “There was a time when the fires occurred behind a school. There was so much dust everywhere. All of this harms them [children]”.

### 3.3. Childhood Chronic Health Conditions

Caregivers explained how daily exposure to the Salton Sea environment contributed to chronic child health conditions, including asthma and other respiratory conditions, as well as allergies and nosebleeds (see Table 3). Many described their children’s lung health as poor—their breathing was weak, and they experienced frequent chest pain and wheezed often. “His throat is closed, he has a whistle”, shared one mother. For many children, their chronic health conditions began at infancy or toddlerhood and continued throughout childhood. A mother shared: “When he was six months old, he got really sick. We took him to Mexicali [Mexico] and they told me to put him on a nebulizer and pat him on the back every 15 min to get rid of all the phlegm”. 

Asthma and respiratory distress. Children commonly experienced an illness trajectory that began with pneumonia, progressed to bronchitis, and ended with an asthma diagnosis or prescription of asthma medications (e.g., albuterol). Throughout this illness trajectory children were repeatedly hospitalized. A caregiver shared:

“The youngest of my children, we constantly have to admit him into the hospital for the same thing: Due to the cold air he would turn purple and would get pneumonia. Sometimes we would go to the doctor and [they] would tell us that he has bronchitis. We have to hospitalize him three or four times a year”.

Chronic and multiple health conditions were common among the caregivers’ children. A mother shared that despite her son having been diagnosed and treated for asthma, he continued to experience chronic respiratory health conditions. 

Allergies. In addition to asthma and respiratory conditions, caregivers described their children as experiencing chronic allergies visible as rashes, irritated eyes, and runny noses. Caregivers talked about their children’s eyes being “watery” or “puffy” especially with heat and humidity. A mother described her son’s coexisting health conditions:

“[He has] respiratory problems and lots of postnasal drip or lots of mucus, like green [mucus]. As if he has the flu, it’s a very strong flu and [he has] watery eyes. His eyes get very puffy and more than anything, it’s when it’s hot, when it’s humid”.

Caregivers understood that air quality and weather conditions affected their children’s health:

“On days when there is a lot of wind, they [children] cannot go outside for very long. They go out for five minutes and come back in, because they cannot stand being outside for long. The air smells bad and their allergies start”.

Nosebleeds. Another common chronic health condition among the caregiver’s children was nosebleeds linked to changing weather patterns and heat. “One of the symptoms my daughter has, and she’s struggled a lot with, is nosebleeds.” This mother explained her daughter gets sudden nosebleeds in which she “cannot stop bleeding” spurred by changes in the weather. Another caregiver shared her understanding of nosebleeds among children in general and the unique case of children living along the Salton Sea:

“I have heard that children get nose bleeds. In my town [in Mexico] they always say that if their [children’s] nose bleeds it’s because of the heat. However, here [along the Salton Sea], it’s all year long that their noses’ bleed. One of my children’s noses bleeds a lot. I think it’s because of the environment, because I’ve already taken him to the doctor and the doctor does not tell me anything. He tells me it’s normal”.

Caregivers discussed their understanding of how the Salton Sea’s environment– air pollution evidenced by sulfuric smells, dust storms, and fires—contributes to nosebleeds. This mother shared: 

“It [son’s chronic health condition] is related to the Salton Sea’s dust. Because my child has a lot of nosebleeds. It is something that is very worrisome. The doctors tell me that there is no medicine to stop the bleeding. I have noticed that in the month of February, this is when my son’s nose bleeds the most. I’ve already taken him to the doctor: ‘Why does my child have a nosebleed in the seasons when it’s windy?’ When we went to the Central Valley, they did not have nosebleeds. Nor did my little girl who has asthma have breathing problems or asthma attacks”.

This quote illustrates a common pattern: When caregivers remove their children from the Salton Sea’s environment, their symptoms improve. As she and others explained, children’s respiratory symptoms reduce or stop altogether, and they no longer experience chronic nosebleeds.

## 4. Discussion

Our study presents caregivers’ understanding of the environmental impacts of the Salton Sea environment on child health. Participants in our study talked about how agricultural practices such as the waste of pesticides being dumped into the lake, the burning of trash on tribal lands, and the lack of water going into the Salton Sea create a highly toxic environment that is harmful to children’s health. Caregivers overwhelmingly report that the Salton Sea environment, including toxic smells and dust storms, contributes to their children’s health conditions. The air quality around the Salton Sea, heavily influenced by the deterioration of the surrounding ecosystem and the blowing of dust particles from the Salton Sea’s drying lakebed, is a significant local, regional, and statewide concern [22,24]. One that is ever more concerning given climate change. Rising temperatures brought on by climate change have contributed to water evaporation, exposing toxins in the lakebed [38]. As our study finds, caregivers’ children are exposed to toxins in the lakebed via dust storms as they breathe this air, which harms their health by contributing to asthma severity (e.g., evidenced by emergency room use and hospitalizations), allergies (e.g., irritated and watery eyes), and nosebleeds. 

Findings from our study contribute to ongoing discussions about the effects of drying saline lakebeds on children’s health [39], providing a critical understanding of such effects on the respiratory health of structurally vulnerable child populations, that is, low-income racial/ethnic and Indigenous Mexican children in rural borderland communities. Study findings are similar to those of Farzan et al.’s [14] research in the Imperial Valley along the southern part of the Salton Sea. They reported asthmatic and non-asthmatic symptoms in children, including wheezing (35%), allergies (36%), bronchitis-like symptoms (28%), and dry cough (33%)—symptoms that are also common among children living along the northern part of the Salton Sea. Our study thus advances the understanding of the effects of environmental exposures on children’s health on both ends of the Salton Sea. It also provides evidence of chronic health conditions that co-present, allergies and nosebleeds, with asthma/respiratory distress in this child population.

A unique finding and important contribution to the literature was the presence of chronic nosebleeds among children living near the Sea. Nosebleeds in children (or pediatric epistaxis) are caused by broken blood vessels and the bleeding of tissues inside the nose. Fluctuations in temperature, humidity, and air pollution are linked to the incidence of epistaxis. Research indicates that the main cause of nosebleeds are high temperatures and low humidity, as well as PM_10_, such as dust, pollen, and mold. Akdoğan and colleagues [40] found in their study among children accessing outpatient care for nosebleeds that epistaxis is positively associated with average daily temperature, and the difference between the maximum and minimum daily temperature is negatively associated with fluctuations in average daily humidity. Kim and colleagues [41] found that air quality or meteorological factors, specifically PM_10_ concentration, were associated with daily epistaxis presentation in both child and adult patient populations in Korea. 

Based on the perspectives of caregivers in our study, epistaxis or nosebleeds in children living along the Salton Sea, is likely related to high temperatures and low humidity as nosebleeds commonly aligned with seasonal temperature changes and were pronounced during the hot summer months of June, July, and August. Yet, there is not sufficient evidence to conclude that temperature fluctuations or fine particulates are the main cause of nosebleeds in children. Our findings and those of others raise questions about the potential impact of climate and seasonal weather patterns and PM_10_ on the health and wellbeing of children living near the Salton Sea. 

## 5. Limitations

Study findings offer insight into the lived experiences of Latinx and Indigenous Mexican caregivers of children with asthma/respiratory problems and co-presenting chronic health conditions of allergies and nosebleeds. The following limitations should be considered when interpreting the findings. First, we merged one-on-one interview data with focus group data to accommodate the needs of study participants who had limited access to digital technology or felt uncomfortable in group settings. While we used the same semi-structured interview guide for both data collection methods, the purpose and goals of each method of data collection differ. One-on-one interviews are best used to obtain individual level experiences and perspectives, whereas focus groups are ideal for obtaining shared collective experiences and perspectives of a community or group [42,43]. For some, it was their first time participating in research, and they did not feel comfortable sharing their perspectives in a group setting, whereas others preferred participating in the research in their native tongue Purépecha both of which limited focus group participation and increased participation in one-on-one interviews. Additionally, in several cases, the participants had limited skills in using digital technology or had limited access to WIFI, resulting in a preferred method of one-on-one phone interviews. For this reason, the focus group sizes were quite small as the ideal focus group size is six to ten participants, which permits facilitators to engage diverse voices and perspectives in group conversations [44].

## 6. Conclusions

Too often, low-income immigrant and minority children in the US live in environments where they breathe highly polluted air [45,46]. This is evident in our study amongst the Latinx and Purépecha immigrant children and caregivers living along the Salton Sea—these families are surrounded by a drying lakebed that emits sulfuric smells and exposes toxic playa that is transported into the air, which children then breathe. This study has important public health implications for vulnerable child populations. The case of the Salton Sea and its effects on the children and families living along its border offers a preview into what is to come in the next several decades. Experts in the field and global health organizations (e.g., World Health Organization) anticipate significant increases in global emissions and air pollutants due to climate change contributing to poor air quality and subsequent increases in asthma and related respiratory conditions [47,48]. Without intervention, structurally vulnerable child populations, like those in our study, will be especially vulnerable to respiratory health consequences of climate change and the effects of poor air quality on health.

## Figures and Tables

**Figure 1 ijerph-20-06023-f001:**
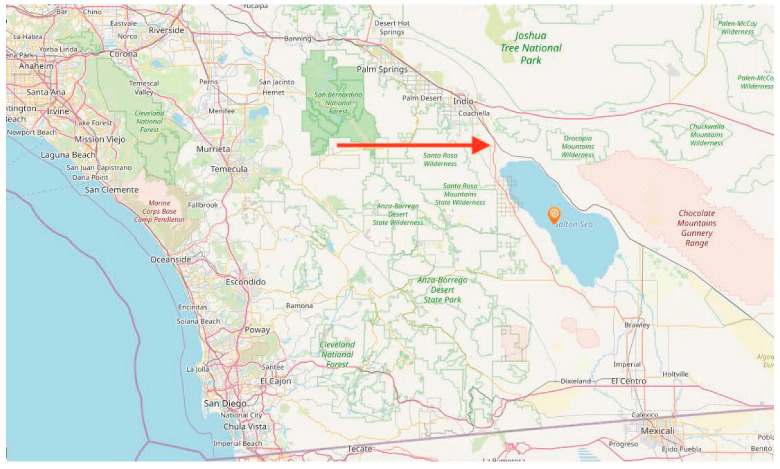
Study Setting along the Salton Sea and its Location in the US-Mexico Borderlands.

**Table 1 ijerph-20-06023-t001:** Characteristics of caregivers and their children living along the Salton Sea (N = 33 ^1^).

Demographic Information	N (%)
Gender (caregiver)	
Female	28 (84.9)
Male	5 (15.1)
Age	
18 to 24	1 (3.0)
25 to 34	5 (15.2)
35 to 44	18 (54.5)
45–55	9 (27.3)
Birthplace	
Mexico	32 (97.0)
United States	1 (3.0)
Ethnicity/Race	
Hispanic or Latino	25 (75.8)
Purépecha	8 (24.2)
Primary language	
Spanish	20 (60.6)
Purépecha	5 (15.1)
Bilingual Spanish/English	3 (9.1)
Bilingual Spanish/Purépecha	5 (15.2)
Education (level completed)	
Never attended school	5 (15.2)
Primary school ^2^	7 (21.2)
Secondary school ^2^	10 (30.3)
High school/GED	9 (27.3)
College/university degree	2 (6.0)
Marital status	
Married or civil union	31 (93.9)
Single or separated	2 (6.1)
Number of children	
1 child	2 (6.1)
2 children	11 (33.3)
3 children	9 (27.3)
4 children	4 (12.1)
5 or more children	7 (21.2)
Employment status	
Employed part time	17 (51.5)
Not employed ^3^	15 (45.5)
Disabled	1 (3.0)
Ever worked as farmworker	18 (54.5)
Type of home	
Apartment	3 (9.1)
Single-family home	10 (30.3)
Trailer	20 (60.6)
Relationship to child	
Mother	28 (84.8)
Father	4 (12.1)
Grandmother	1 (3.1)
Focal child place of birth	
United States	31 (93.9)
Mexico	2 (6.1)
Focal child primary language	
English	6 (18.2)
Spanish	4 (12.1)
Bilingual English/Spanish	22 (66.7)
Trilingual English/Spanish/Purépecha	1 (3.0)
Focal child age	
0 to 5	2 (6.1)
6 to 11	9 (27.3)
12 to 14	10 (36.4)
15 to 18	12 (45.5)
Caregivers’ perception of focal child’s overall health	
Very good	1 (3.0)
Good	15 (45.5)
Moderate	15 (45.5)
Very bad	1 (3.0)
Unsure	1 (3.0)

^1^ A total of 36 people participated in either a focus group or one-on-one interview; only 33 participants completed the socio-demographic survey. Some numbers and percentages may not add up due to missing data. ^2^ In Mexico the school system differs from that of the US education system. Primary school includes kindergarten to 6th grade, secondary school 7th to 9th grade, and the second stage of secondary school includes 10th to 12th grade. ^3^ Many of the participants who indicated they were not employed in the survey indicated in the qualitative interviews that they were stay-at-home mothers.

**Table 2 ijerph-20-06023-t002:** The Salton Sea Environment.

Theme	Caregivers’ Perspectives
Toxic smells	“It [the sea] does affect them, because in this community where I live the smell is more frequent and stronger. My daughter who has asthma tells me that she feels like she’s choking when she smells that odor. My daughter tells me I feel really sick, my chest hurts when the sea smells like that.”~Purépecha interview participant, female
	“Every time you smell it [sulfur] a lot, there goes the pollution there … smells the smell here. Sometimes he’s [child] even incapable of breathing, the same smell too.”~Latinx focus group participant, female
	“… that smell is something that even irritates children’s eyes. Oh, my little girl is very irritated.”~Latinx focus group participant, female
	“The vapor of the lake [Salton Sea], the smell, I think it has a lot to do with that vapor during the time of, of heat, because it evaporates all of that, that smell, that humidity. And, I have seen that ultimately yes, the biggest [smell] has happened during times of heat.”~Latinx focus group participant, female
Dust storms	“When it is very windy and there is a lot of dust, the chances of having a [respiratory] crisis is higher, either from allergies or, when, it’s very difficult to breathe [because] the [the air] is mostly dust.”
	“The children are outside a lot and there is a lot of wind, and the wind brings the dust from the lake, which is drying up.”~Purépecha interview participant, male
	“It’s [the Salton Sea] is drying up. There is a lot of wind … it’s not something that is just once in a while, but it’s constant. A lot of dust comes from there [the Salton Sea], all of that harms a child who has asthma. Everything, the dust, the smells that come off the lake.”~Latinx interview participant, female
Agricultural chemical exposure	“The ranchers have been throwing the chemical waste in the lake, that is why she [participant] says that the dust is associated with public health.” ~Latinx and Purépecha focus group participant, female
	“All the chemicals thrown in the fields … it goes into the lake. I mean, this is something that is notorious … when you go to the lake and you see the waste from the pesticides in the water, in the dirt …”~Latinx focus group participant, female
	“The pesticide, the chemical wastes of the pesticides, they throw it here [Salton Sea] in the water. And, well also, the lake is drying up … it’s the children who are breathing it … which is perhaps why children have asthma.”~Purépecha interview participant, male
Fires	“The fires started; the smoke from the fires came all the way here [Salton City].”~Latinx focus group participant, male
	“I see that there was a time where the fires were here behind the school, there was a lot of smoke everywhere, all of this hurts them [children].”~Latinx focus group participant, male
	“You can’t go out if it’s windy. For example, left now, that there are fires, it’s even worse, the symptoms [of h child] are getting worse … so it’s really alarming that kids can’t, sometimes, go out. When they go to school, to be able to go out to be with their classmates, [they] have to go [back inside] because they have problems because they cannot do the exercises that are required of them at school and they have to leave them aside because they have severe allergies or asthma, and that they [kids with asthma] always have to have their inhalers in their backpacks or at school.”~Latinx focus group participant, female

**Table 3 ijerph-20-06023-t003:** Children’s Asthma and Co-presenting Health Conditions.

Theme	Children’s Lived Experience as Shared by Their Caregivers
Asthma	“The youngest of my children, every so often we have to admit him to the hospital for the same reason … he would catch pneumonia and sometimes we would go to the doctor and he [the doctor] would say that he [the child] had bronchitis and almost always, for almost a year we have to hospitalize him three times, four times a year.”
	“… [child] has more problems with asthma attacks in the month of February, March, April, and June.”~Purépecha interview participant, female
	“When he was six months old, well, he got really sick. We took him to Mexicali, and they told me to put a nebulizer on him and pat him on the back every 15 min to get rid of all the phlegm.”~Latinx interview participant, female
	“… Ever since they operated on him, now, because he used a lot of those little devices to breathe, they put tubes like those in his nose, like little hoses with the mask, he had to use the tubes every so often and a little device that made steam like that to cleanse the lungs, and they gave him an inhaler, they gave him one or two in case he ran out of one, so he had a replacement…. The doctor had said it was like asthma that [my] kid was already having asthma, and they sent him to a specialist, and they had to operate on him there [hospital] …”~Purépecha interview participant, female
	“… he [child] already visited a specialist and checked him out and already said that he had to burn his tonsils with a laser and remove the meatiness [flesh] from his nose. And, well, thanks to that, since then, he has never gotten sick again, not the flu nor any coughing. Because he started coughing and coughing and you could almost see him, he was going to die when he started coughing. He couldn’t breathe because he was drowning and what was happening to him scared us at night, when he started snoring … it scared us because he couldn’t breathe. We had to take him to the emergency room and his tummy was jumping a lot, because he couldn’t breathe, his tummy was jumping like he was breathing very fast, but at the same time he couldn’t let it go again and his tummy was jumping very fast.”~Latinx interview participant, male
Allergies	“My eldest child gets them [allergies], that’s when he gets his attacks … and he has an inhaler. But he’s almost like, every day, at night I hear him breathing, even though he’s not using the inhaler, I hear him struggle …. I don’t sleep much because I’m waiting to see if he’s breathing well or not.”~Latinx interview participant, female
	“They [doctor] diagnosed [child], [with] asthma, because she [doctor] did some different studies, she sent me to do tests, the pulmonologist, allergies, everything … [doctor] sent him [child] to him [doctor], the nose and throat specialist because he [child] was bleeding from the nose. Then the doctor told him that he had a lot of allergies and that his veins were wearing out. That he [child] had a deviated septum and that his veins were so irritated, that they [doctors] could burn them.”~Latinx interview participant, female
	“… there are respiratory problems and a lot of runny noses, or a lot of mucus, like green. As if [child] had a really bad cold, and [child’s] eyes are watery, their [child’s] eyes swell. And, this is mostly [happening] when it’s hot, when it is humid”~Latinx interview participant, female
Nosebleeds	“There are seasons where they [kids] suddenly bleed. My child … sometimes his nose bleeds or he gets up and there is blood on his pillow. So yeah, if it’s something like that, [I have] some concern. He would even stain his feet, so it was like a little stream of blood that would spill.”~Latinx interview participant, female
	“One of the symptoms that my daughter has, and has struggled a lot with it, is the bleeding. Every change in weather there is a tremendous amount of [nose] bleeding that she suddenly just goes down and can’t stop bleeding”~Latinx focus group participant, female
	“Other children that I know well who have nosebleeds, and it is something that they [kids] don’t need to be out in the sun that much to start bleeding and bleeding and bleeding …”~Latinx focus group participant, female

## Data Availability

The datasets used and analyzed for this study can be available from the corresponding author upon reasonable request.

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
