# Peer review of "Latinx and Indigenous Mexican Caregivers’ Perspectives of the Salton Sea Environment on Children’s Asthma, Respiratory Health, and Co-Presenting Health Conditions"

_ijerph, 2023, doi:10.3390/ijerph20116023_

Round 1

Reviewer 1 Report

This is a very interesting study that contributes to the body of literature surrounding the environmental health and justice issues associated with the Salton Sea.  The qualitative research provides insights of a disadvantaged population experiencing increased negative health outcomes due to exposures to environmental hazards.  However, the organization of the manuscript and the presentation of the methods, data collection/analysis and results are difficult to follow.   I believe this paper has merit and with revisions, can be an important contributor to this issue.

1.  Throughout the paper, Salton Sea and Sea are used interchangeably, however, there is no indication that Sea is an acronym.  For easier reading, I recommend either using Salton Sea or Salton Sea (Sea) and not both.

2.  The abstract states that the caregivers were selected based on children with asthma or respiratory distress.  The introduction focuses on pediatric asthma cases.  The methods section recruits parents onto an advisory board with children diagnosed with asthma.  The setting focuses on childhood asthma.  The participant recruitment is based on asthma symptoms (which are not explicitly stated).  In the population characteristic section, the focal children were a child with asthma or respiratory distress.  In the results section, allergies were explored.  In the discussion, there was great detail about the pediatric epistaxis or nosebleed findings.  If this paper is about asthma and respiratory distress, more information is needed in the introduction to explain childhood respiratory distress.  Then, if the recruitment of participants in the study are caregivers of children with asthma symptoms, you need to describe why children with respiratory distress were excluded from your sample.  Or, is this a study that focuses on childhood asthma?

3.  Throughout the manuscript, you interchangeably use PM2.5, PM10, dust, aerosolized contaminants/aerosols, toxic dust, dust storms etc.  Air quality and ozone are also included.  From an environmental hazard and exposure standpoint, these represent different hazards and exposures.  For example, are you talking about the U.S. National Ambient Air Quality Standards (NAAQS) for PM2.5 and ozone?  NAAQS also includes lead, NOx, SOx and carbon monoxide which can also be classified as toxic.  There should also be distinctions between dust and aerosols.  Please describe hazardous exposures within environmental/occupational terminology and discern between ambient and toxic air constituents.

4.  The use of "global" is only used once in line 375 throughout the paper until the conclusion that focuses heavily on global air quality and negative health outcomes.  The conclusion and the other sections of the paper are disjointed and should be reconciled with a global emphasis throughout the paper or reworking the conclusion to match your research.

5.  Method sections 2.1 - 2.1.5 do not provide a clear understanding of the study.  In section 2.1, I am unclear why adult health inequity issues are a part of this paper since this research is on childhood respiratory disease.  Section 2.1.1 belongs in the introduction section.  Section 2.1.2 refers to vulnerable populations.  Social vulnerability and the variables that define this population are not the same as disadvantaged populations that are defined by indicators within the different categories of social determinants of health.  Structural, social, and political health inequities are not proper terminology.  Social determinants of health are the categories of neighborhood constructs that contribute to health inequities that can include social cohesion, institutional level policies such as structural racism, and political capital with voting rights.  I recommend visiting the US Healthy People 2030 Social Determinants of Health website to present health inequities within their definitions.  As stated before, in section 2.1.3 definition of recruitment needs to be parsed out more with the explanation of why children with asthma symptoms were the target audience.  In section 2.1.4, you describe a survey with no additional information about the tool.  Was this survey validated?  If so explain.  Did you use questions from a study that had a validated survey tool?  How many participants in each of the focus groups?  In section 2.1.5, what software did you use for coding and thematic categories?  You also describe using a matrix.  The completed matrix would be very helpful as a figure to understand your results.

6.  In the results section 3.1, the paragraph and the table are redundant.  A reader can review the table for the results without reiterating these outcomes in writing.  If something is not apparent in the table, then, describe it in the paragraph.

7.  In section 3.2, the title, "An Overview:  The Salton Sea Environment is confusing.   Lines 213 - 222 should be in the introduction.  Is toxic smells and windstorms main themes in your matrix?  In these two paragraphs, the first couple of sentence seem appropriate in the introduction because they explain the issue.  The quotes are getting lost and there are no pseudonyms attached to relate it specifically to a participant.  In section 3.3, I highly recommend looking at ways to present quotes within your citation style.  The current layout is very difficult to read and follow because it is interwoven with facts, commentary and quotes.

8.  Citations are missing throughout the paper, for example, lines 341 - 350, 387 - 389, 213 - 222 etc.  Please review the paper to ensure information is cited appropriately.

9.  The references to COVID-19 in the paper and why it is included in the survey is not clearly explained.  What is the connection to COVID and the hazardous exposures contributing to respiratory disease in children?

I think that this paper has great importance by including community-based participatory research examining an environmental health and justice issue negatively impacting minority children.  Often times, environmental justice research investigates risk, hazards, and exposures (which is important) without enough emphasis on community members.  With revisions, I believe this paper fills an important gap.

Author Response

Please see attached response.

Reviewer 2 Report

Reviewer’s report

Title: Latinx and Indigenous Mexican Caregivers’ Perspectives of the Salton Sea Environment on Chronic Childhood Health Conditions

Overall: Thank you for the opportunity to review this manuscript. It is a very interesting and important topic and I enjoyed reading it.

Abstract: Give some more information on the Salton sea (from your method 2.1.1. The Salton Sea) and how it’s contributing to chronic health conditions (i.e. dust concentrations and other toxins).

Introduction:

Line 45-48: there are many more sources of PM2.5 including natural sources. I suggest rewording this section to make it clear your list is not inclusive off all sources. E.g. ……of >2.5 microns (PM2.5); a form of atmospheric particulate matter from sources such as wood burning, fossil fuels, and heating oil or coal, among others.

Methods:

2.1.2. Structurally Vulnerable Populations Living Along the Sea

·         Just curious what is the percentage of Latinx and Indigenous Mexican population in this region? 

Line 165-167: Did all invited participates have access/skills to use zoom and the online survey? Was there any exclusion or non-participation due to lack of internet connectivity or technological skills to use these methods?

Results:

Table 1: Please note make it clear that gender refers to caregiver in the table.

Footnote 2 of table 1: I am slightly unsure of this comment. You only have 1 person in your table born in the US, so does this mean virtually all only completed primary/secondary school? You have 7 listed as high school/college?

Lines 212-222: You need to provide references for these assertations. This applies to your opening statements for windstorms and other sections too.

Lines 223-223: Are these toxic smells noted by the researchers or past research/survey work? This needs to be made clear (and perhaps cited)

Lines 243-247: Is this a participant quote? Please include quote marks.

Line 270: Please give mor clarity as to what [in his ?] refers to – why is there missing data here?

Line 320-323: I do worry about some of the language used in this manuscript. The authors appear to be making causal statements that definitely link the respiratory outcomes to the Salton Sea. Unless they can support with references, then they should make it clear the respondents believe it is linked. While I agree there is a compelling case here, as environmental scientists we must maintain our unbiased language and accurately report our results. This should be addressed throughout the manuscript.

Discussion:

Line 341: Please support this strong assertation with references.

The first paragraph needs to be referenced and the strength of the language reviewed.

Line 351-354: This is new information and should not be presented in the discussion section. Please also support with some figures if possible – how many caregivers reported returning to health care for the same issues repeatedly?

Line 365: Again, this is new information that should be included in results with more supporting information.

Reviewer 3 Report

The manuscript aims to investigate the impact of the Salton Sea environment on the health and well-being of socially and economically disadvantaged children living along its borders. To meet the objective, qualitative research was carried out that included 4 focus groups, interviews with 32 caregivers, and 26 completed sociodemographic surveys. One of the novelties of this study focuses on the community's understanding of the Sea's impact on child health through qualitative research with Latinx and Indigenous Mexican caregivers of children with asthma symptoms. In this sense, the manuscript is relevant to discuss the usefulness of qualitative research to examine the link between public health and the quality of the environment, in contrast to only using quantitative research.

Here, I wish to provide some suggestions/questions that I hope to help authors to make this interesting work clearer:

A location map is needed to understand the problems described in the manuscript that refers to the border between the United States and Mexico and the Salton Sea.

A more extensive review/description of the environmental conditions of the Salton Sea is needed. 

Although the sample is small, the limitations of obtaining a larger sample are explained. Otherwise, it would take a longer time to get more responses. However, is there any possibility of carrying out fieldwork with a larger sample? 

There is a lack of data on the economic aspects of the interviewees/people who offered their testimonials (for example, their average monthly economic income). This data is relevant to understand the socioeconomic conditions of the sample.

I would like to know what is the difference between this manuscript and the manuscript entitled "The Salton Sea Environment and Child Health: Latinx and Indigenous Mexican Caregivers' Experiences of Caring for Children with Asthma, Allergies, and Nosebleeds". Apparently, this last manuscript has not been published and is under review in Environ. Health Perspect. 2022.

Author Response

Please see attached response. 

Round 2

Reviewer 2 Report

Reviewer’s report

Title: Latinx and Indigenous Mexican Caregivers’ Perspectives of the Salton Sea Environment on Children’s Asthma, Respiratory Health, and Co-presenting Health Conditions

Overall: Thank you for the opportunity to re-review this manuscript. The manuscript is much improved by the revisions and is a really wonderful piece of work. I am now invested in the authors work and look forward to seeing more work on this topic. A few minor items need addressing before publication.

Abstract: Perhaps mention the pollution link to the Salton Sea (i.e. dust concentrations and other toxins).

Keywords: ‘Childhood asthma and co-presenting health conditions’ will not index well. Change to ‘asthma’ ‘child health’ and ‘respiratory conditions’.

Introduction: Line 48-52: Fine particulate matter should be less than (< 2.5 microns)?  

Results:

Line 236 – Revise this sentence. Incomplete and spelling error (“The focial children were between the ages of”)

Line 242 – footnote of table 1 was inconsistent with line 231. Was it 36 caregivers participated in the study and 33 completed the survey?

Line 252: spelling error in ‘health’

Line: 347-348: Advise revising to: ‘air pollution evidenced by sulfuric smells’, as it is not the smell that causes the pollution.

Table 2 – Quote marks for this first perspective for toxic smell

Discussion:

Line 426 – spelling error “participating”

Line 366: I still have concerns with the strength of the language here. Your study provides a compelling narrative around this issue, but it does not provide ‘evidence’ in its strictest sense. Perhaps add a qualifier – Our study present compelling community reports of the environmental… OR Our study presents on communities experiences of environmental harms from the…

I would also suggest softening the language on line 370 – perhaps caregivers overwhelmingly report that the Salton Seas environment.

Author Response

Overall: Thank you for the opportunity to re-review this manuscript. The manuscript is much improved by the revisions and is a really wonderful piece of work. I am now invested in the authors work and look forward to seeing more work on this topic. A few minor items need addressing before publication.

  • Response: Thank you. We very much appreciated your thoughtful feedback.

Abstract: Perhaps mention the pollution link to the Salton Sea (i.e. dust concentrations and other toxins).

  • Response: We included reference to dust concentrations and other toxins in the first sentence of the abstract.

Keywords: ‘Childhood asthma and co-presenting health conditions’ will not index well. Change to ‘asthma’ ‘child health’ and ‘respiratory conditions’.

  • Response: Thank you for this feedback, we revised the key words accordingly.

Introduction: Line 48-52: Fine particulate matter should be less than (< 2.5 microns)?  

  • Response: We revised this to ‘2.5 microns or less’.

Results: 

Line 236 – Revise this sentence. Incomplete and spelling error (“The focial children were between the ages of”)

  • Response: We omitted this sentence from the manuscript body as the information is in the table.

Line 242 – footnote of table 1 was inconsistent with line 231. Was it 36 caregivers participated in the study and 33 completed the survey?

  • Response: we updated the footnote to 36 completed qualitative interviews and 33 completed surveys.

Line 252: spelling error in ‘health’

  • Response: We corrected this error.

Line: 347-348: Advise revising to: ‘air pollution evidenced by sulfuric smells’, as it is not the smell that causes the pollution.

  • Response: We revised this sentence accordingly.

Table 2 – Quote marks for this first perspective for toxic smell

  • Response: We added quotation marks here.

Discussion:

Line 426 – spelling error “participating”

  • Response: We corrected this spelling error.

Line 366: I still have concerns with the strength of the language here. Your study provides a compelling narrative around this issue, but it does not provide ‘evidence’ in its strictest sense. Perhaps add a qualifier – Our study present compelling community reports of the environmental… OR Our study presents on communities experiences of environmental harms from the…

  • Response: We updated to this ‘Our study presents caregivers’ understandings of the …’

I would also suggest softening the language on line 370 – perhaps caregivers overwhelmingly report that the Salton Seas environment.

  • Response: We updated this sentence accordingly.

Reviewer 3 Report

I could not see the map or figure 1.

Author Response

Reviewer comment: I could not see the map or figure 1.

Our response: We have re-uploaded Figure 1, the map of the US-Mexico border that illustrates the location of the Salton Sea to the manuscript system. We also placed this in the manuscript and labeled the figure: Figure 1. The Study Setting along the Salton Sea and its Location in the US-Mexico Borderlands.

Round 3

Reviewer 3 Report

I have no more comments.

Author Response

Thank you for your positive feedback.